# Precision Low-Cost Compact Micro-Displacement Sensors Based on a New Arrangement of Cascaded Levers with Flexural Joints

**DOI:** 10.3390/s23010326

**Published:** 2022-12-28

**Authors:** Che-Chih Tsao, Yi-Chun Tseng, Yu-Sheng Chen, Wei-Hsuan Chang, Li-Ting Huo

**Affiliations:** Department of Power Mechanical Engineering, National Tsing Hua University, Hsinchu 30013, Taiwan

**Keywords:** displacement sensor, flexure, micro-displacement, Hall sensor

## Abstract

Existing displacement sensors of micrometers to sub-micron precision are expensive and have various limitations. This paper reports the design and development of a new contact type compact micro-displacement sensor of sub-micron precision for a fraction of the cost of commercial devices. The basic concept of the new sensor system applies a mechanical magnifying mechanism to magnify a displacement at sub-micron to micron level and uses a low-cost Hall sensor to measure the magnified displacement. Various conceptual designs for the mechanical magnifying mechanism based on cascaded levers with flexural joints were studied and a final design, featuring side-by-side placement of lever structures in a multi-planar layout with adjacent levers coupled by L-shaped coupling foils, was devised. Prototypes of two different sizes and constructions with mechanical magnification ratios over 100 were made and tested. Measurement repeatability and accuracy to sub-micrometer level and a resolution down to hundredths of a micrometer were demonstrated by a compact Alpha Model prototype. Design modification of parts and a corresponding small lot size production procedure were devised to provide an estimated bill of material cost per unit under US$100.

## 1. Introduction

### 1.1. Problems and Motivations

Precision displacement sensors or gauges are useful in measuring and checking dimensions of manufactured parts, positions of machine stages, assembly precisions of components in a machine system and in other areas. However, for in-process, in-machine precision measurements at micron to submicron level, there is currently no low-cost and compact choice. Traditional mechanical displacement gauges, such as dial indicators or meters based on gears and spiral spring, have a precision of about 2 μm (micron, or micrometer) and are not suitable for sub-micron level measurements. Highly precise displacement measurement systems, such as interferometers, can easily make sub-micron measurements, but they are extremely expensive and bulky and generally not suitable for in-process or in-machine applications. For in-process, in-machine precision measurements, several commercial devices are available, as shown in Table 1. The list includes sensors of non-contact type and contact type. Capacitive sensor, eddy current sensor, laser triangulation and color confocal sensor are non-contact type. LVDT (including its probe part) and digital contact displacement sensor are contact type. For micrometer level measurement precision, capacitive sensor, eddy current sensor, digital contact sensor and color confocal sensor can reach one micron accuracy and sub-micron resolution. However, they are still expensive. Table 1 shows prices of single sensor heads in USD, not including their companion electronic boxes. The Hall sensor is very cheap, but its precision is over several micrometers.

In addition to costs, commercial devices also have various limitations. While non-contact measurement is appealing, capacitive sensors have a very small measurement range. Eddy current sensors can only be used on good conductors [7]. (Chap. 3) Sensors based on laser triangulation have good resolution but limited linearity [4]. Additionally, the reflective property of a surface always affects the sensing of any optical sensor. Contact type sensors can be applied, in general, to any rigid surface. Existing contact type displacement probes, such as the Keyence example listed in Table 1 and many other commercial models, generally use an axially moving long rod as the probe, together with an opto-electronic sensing unit to “read out” the displacement of the rod. This design requires size and carries intrinsic cost.

Accordingly, if a displacement sensor can be made to have a compact size, to have a precision comparable to existing precision sensors, and to cost only a fraction of or even an order of magnitude less than those commercial sensors, then it can be applied in many situations in large quantities in the field. For example, a machine tool or a positioning stage can have several sensors installed at strategic locations and combined measurement results can be used to track machine deformations or position errors in multiple directions due to thermal or load effects in real time. Such information can lead to further improvement of process accuracy. These small devices can also be applied to factory machines for maintenance monitoring and in production lines for process monitoring, all at costs lower than existing choices.

This paper reports the design and development of a new contact type precision micro-displacement sensor for measuring a micro-displacement of a target surface with a measurement range from a few micrometers to hundreds of micrometers and a measurement resolution on the order of 0.01 to 1 micrometer and a measurement accuracy on sub-micron to micrometer level. The basic concept is to apply a mechanical mechanism to magnify a sub-micron displacement to be measured by about 100 times so that the magnified displacement can be measured by using an existing low-cost displacement sensor, such as a Hall sensor. That is, with a reliable mechanical magnifying mechanism of consistent and repeatable magnification ratio, a low-cost displacement sensor can measure a micro-displacement to about 1/100th of its own resolution and accuracy.

### 1.2. Technical Background in Displacement Ampilification Mechanisms

Displacement amplification mechanisms can be found in industrial applications and in academic research. In the broadest sense, displacement amplification or magnification mechanism can be separated into two categories: Integral structures and non-integral structures. By integral structures we mean structures fabricated from a monolithic piece of solid material or an assembly of parts joined into an integral solid structure. Between any parts of an integral structure, there is no mechanical contact with relative motion. On the other hand, non-integral structures include discrete parts that are in contact, but not joined into one solid, and there can be relative motion between parts in mechanical contact. Kelemenová [8] summarized amplifying mechanisms of various types, including both integral and non-integral structures.

Flexure-based mechanisms belong to the category of integral structures. In general, because there is no mechanical contact with relative motion and thereby no need of processes of hardening or polishing of contact surfaces, flexure-based mechanisms should have better precision and repeatability and potentially a lower cost compared to mechanisms of non-integral structure. However, most implementations seen in industry or in academic publications are case by case and have very application-specific designs, not suitable for development into low-cost off-the-shelf products. Some examples are briefly reviewed below. Chen et al. [9] provided an extensive review of flexure-based displacement amplification mechanisms. One frequently applied flexure-based mechanism is the bridge structure, such as examples shown in Juuti et al. [10]. Bridge structures can be modified and stacked into a honeycomb link, as described in Muraoka et al. [11]. Another frequently applied mechanism is the lever structure, from single to multiple stages. For example, Henmi et al. [12] describes a two-stage lever amplifier used in an actuator. Most examples described in the literature referenced above are for applications to amplify motions of actuators, rather than for displacement measurement, although the principle of the mechanisms can be applied in both ways. In Yu et al. [13], a two-stage lever structure is applied to reduce a displacement from an actuator to obtain micro-scale displacement for thin film measurement applications. Yamauchi [14] described a device for measuring rock deformation in a drilling tube applying a single stage lever mechanism. Bae et al. [15] described a piezoelectric actuator driven micro-stage with a build-in magnifying mechanism for displacement measurement. The magnifying mechanism was a two-stage lever system with monolithic hinges all made in a piece of metal slab and had a magnification ratio of 30. Such planar integral structures are also seen in MEMS structures, such as a micro-actuator structure reported by Nada et al. [16].

## 2. Basic Concept and Studies of Conceptual Designs

### 2.1. Basic Concept of the Low-Cost Precision Sensor

The basic concept of the contact type precision micro-displacement sensor of this research is summarized in Figure 1, using a magnification ratio of 100 as an example. From Table 1, we see that the Hall sensor has a significantly low cost but a measurement resolution of about 5 μm. It has an accuracy of about 0.1–1% of full scale and a range of 0.25–2.5 mm. That is, for a measurement range on the order of 1000 μm and a resolution/accuracy of 1–5 μm, the Hall sensor is a very cost effective choice. Now, for measuring a displacement down to 0.1 μm, if we can magnify this tiny displacement by a factor of, say, 100, then we will be measuring 10 μm, which falls well within the Hall sensor’s capability. That is, if a magnification mechanism can be devised to reliably magnify sub-micron displacements with good repeatability, then a Hall sensor can be used to resolve a micro-displacement of 0.1 μm with a full range of 10 μm, while the Hall sensor itself is measuring displacements in a full range of 1 mm with a resolution of 10 μm.

Further, if such a mechanical micro-displacement magnifying mechanism (MDMM) can be made small with low cost, then a precision Low-cost Compact Micro-Displacement Sensor (LCMDS) can be constructed by combining it with a Hall sensor, which is also cheap and of small size.

By applying different mechanical magnification ratios, the micro-displacement sensor can have different measurement ranges, as shown in Table 2. For a specific sensor measurement range, the corresponding best measurement accuracy and resolution can be estimated by shrinking the general resolution and accuracy of a typical Hall sensor by the magnification ratio.

As pointed out already, because there is no mechanical contact with relative motion, flexure-based mechanisms should have better repeatability and lower cost compared to mechanisms of non-integral structure. There are two frequently seen flexure-based mechanisms: Bridge structure and lever structure. The latter is comparatively simple and easy to construct. Accordingly, lever structure was selected for the design and construction of the mechanical displacement magnifying mechanism of this research. To obtain large magnification ratios, larger than 25 or so, multiple lever structures need to be used and connected in cascaded stages into an integral structure of levers. A magnified displacement in a lever structure in one stage drives another lever structure in the next stage and the original displacement is magnified further.

By geometry, the cascaded lever structures can take two kinds of layout arrangements. The lever structures can be disposed with one lever structure on top of another and with the planes of motion of the arms all on the same plane. This can be called the co-planar layout. The lever structures can also be disposed side by side with all arms in parallel and with the plane of motion of each arm different but parallel to each other. This can be called the multi-planar layout.

Flexural joints are used as hinges of the levers and for coupling between lever structures of successive stages. There are two categories of flexural joints, namely monolithic structures, which can be made as integral parts of a monolithic mechanism, and flat-springs, which need to be assembled to a mechanism [12] (Section 8.6).

Most existing flexure-based displacement amplifying mechanisms, as referenced in the literature review above, use co-planar layout, especially when the mechanism is made from a monolithic piece of solid metal with build-in monolithic flexural joints. This could be due to ease of manufacturing because flexural joints in integral forms can be made by drilling holes on a metal slab followed by making lever arms by wire EDM (Electric Discharge Machining) of the slab.

In this research, MDMM conceptual designs of both co-planar layout and multi-planar layout using monolithic and flat-spring flexural joints were explored in order to find a reliable and compact design of cascaded lever structures with flexural joints of a magnification ratio of about 100.

### 2.2. Conceptual Design A: Monolithic Structure of Co-Planar Layout with Monolithic Flexural Joints—Exploratory Study

A monolithic three-stage cascaded lever structures was first designed, as shown in Figure 2, and studied. Each lever structure was designed to have a magnification ratio of five, with the goal of a total magnification ratio over 100. The input contact place is located at A in the first stage lever L1 and the output is located at B on the third stage lever L3. The monolithic flexural hinge structures (LH1, LH2 and LH3) were in the form of circular notch and were designed to avoid fatigue, assuming the application of aluminum, by using the stress formula from Merken [17] with a rough estimate of bending angles. The output end of lever L1 is coupled to the input place of lever L2 by a coupling joint CF12, which is a smaller monolithic flexural hinge. The coupling between stage 2 and stage 3 at CF23 is the same.

Deformation analysis using Ansys software was applied to the structure to observe the displacement of the levers and deformation of the joints and the magnification ratio was found to be very small. Figure 3a is a total deformation graph of the structure when a force of 1 N is applied downward at contact place A, showing that the displacement at output B is only slightly larger than that of input A.

The main issue was found to be at the coupling joints, CF12 and CF23. As shown in a close up view of joint CF23 in Figure 3b, shear strains are obvious. This is because, when two adjacent lever arms rotate about their hinges, even at small angles, relative translational displacement also occurs due to the rotations, thereby causing relative shearing. The small coupling joints under shearing prevent the lever arms from further rotations and therefore limit the magnification effect.

To resolve the issue, additional flexure must be built into the coupling joints to accommodate both relative rotational movements and relative translational movements. To achieve this in a monolithic co-planar layout, design of the coupling joints could become complicated and the overall size of the structure could increase. Another issue of the co-planar layout is that the height of the structure, measured along the direction of displacement at position A, can become too large and no longer “compact”. For these issues, the design approach was switched to multi-planar layouts.

### 2.3. Conceptual Design B: Assembled Structure of Multi-Planar Layout with Monolithic Hinges but Flat-Spring Couplers—Exploratory Study

In this design, levers are disposed side by side with all arms in parallel and with the plane of motion of each arm different but parallel to each other, as the multi-planar layout shown in Figure 4. By this multi-planar layout, the height of the assembly can be reduced.

A challenge associated with this approach is how to couple levers placed side by side to effectively transfer the displacement from one stage to the next stage. An L-shaped flat spring foil was devised for this purpose. As depicted in Figure 4, the coupling foil CF23 is mounted with a horizontal bare section *D*_2_ on top of lever 2 and a vertical bare section *D*_1_ to the side of lever L3. The same is for the coupling foil CF12. As already described, there are both rotational relative motions and linear relative motions when two adjacent lever arms rotate about their hinges. Figure 5 shows an exaggerated depiction of this phenomenon. The horizontal bare section of the coupling foil is devised for accommodating the rotational relative motions and the vertical bare section for linear relative motions.

Key parameters of this design approach were tested and studied by using finite element models, as one example shown in Figure 6a, with the goal to achieve a total magnification ratio over 100. The following observations were obtained.

(1)The coupling foils should be pulled rather than pushed: The first question is in what orientation the L-shaped flat spring foil should be mounted between two adjacent lever arms. In the configuration of Figure 4a, when an input force pushes on position A the output displacement of lever L2 “pulls” lever L3 through the coupling foil CF23; while in the configuration of Figure 4b, lever L2 “pushes” lever L3. Considering mechanics of material, pulling should be preferred because it results in mainly tension, which is easier to take by the coupling foil than pushing, which has a potential of causing bulking in the foil. Finite element simulations using Ansys software confirms this consideration, as a model configuration of Figure 4b gives a smaller magnification ratio than the model configuration of Figure 4a does.(2)For an assembly of lever structures of a fixed size, the thickness of the coupling foils has a minimal effect on the magnification ratio.(3)The coupling foils should be in contact with the lever arms with widths as large as possible, as depicted as *W*_1_ and *W*_2_ in Figure 4a, so that there is sufficient stiffness in the coupling, which helps to improve the magnification ratio.(4)A larger width (as *W*_2_ in Figure 4a) of each lever structure also increases the structure stiffness and thereby helps to improve the magnification ratio. However, activation force also increases with a thicker lever structure. For an input force not exceeding 1 N, the width of the levers in the design of Figure 6a was set to be 8 mm.(5)The sizes of the bare sections of the coupling foil affect achievable magnification ratio significantly, especially the horizontal bare section *D*_2_. The bare sections *D*_1_ and *D*_2_ on the coupling foil are devised to accommodate the relative linear movements and relative rotational movements between adjacent, coupled lever arms. Sizes too small make the cascaded lever structures too stiff to deform, but sizes too large make the coupling too soft to effectively transfer displacement, both hurting the magnification ratio. Therefore, for a cascaded lever structure of a given size, there should be an optimal range of bare section sizes. Different values of *D*_1_ and *D*_2_ were tested to observe their effects on the total magnification ratio and the results are summarized in Figure 7. *D*_1_ = 2 mm and *D*_2_ = 1.6 mm were selected as close to optimal for this design.

After the above adjustments and optimization, the best total magnification was 57, which was still far short of the targeted 100. Close examinations of the deformation at different stages and locations in the model of the cascaded lever structures revealed that there were significant shear strains in the horizontally placed monolithic flexural hinge, as shown in Figure 6b, and these shear strains ate up a significant portion of the ideal magnification ratio. Thus, the hinges needed improvement, which led to the next conceptual design.

### 2.4. Conceptual Design C: Assembled Structure of Multi-Planar Layout with Flat-Spring Couplers but Re-Oriented Monolithic Hinges—Exploratory Study

In order to avoid shear strains in the hinges, the horizontally placed hinges should be reoriented to become aligned along the direction of input force, that is, in the vertical direction. The vertically placed monolithic hinges can be arranged to respond to the input force by tension or by compression. From observations obtained in Section 2.3, flexural joints in tension may be a better choice. However, a tensile arrangement of monolithic hinges in the cascaded lever structures could result in a somewhat complicated design. This is because, in order to load the hinge in tension, the hinge needs to be placed above the lever arm, and since a monolithic hinge takes up certain space, the distance between the axis of the hinge and the input place (effort point) cannot be too small. This will not be helpful for designing a “compact” sensor device. As a result, we decided to test a compressive arrangement of monolithic hinges first.

Finite element modeling was applied to help determine the optimal thickness of the flexural section of the hinge in each stage that minimizes deformation in that single stage. Other parameters were kept the same as those of the conceptual design B. Simulations of total deformations indicated that a total magnification over 100 is achievable. A detailed design was made, as shown in Figure 8. Finite element simulation showed that a downward force of 5 N at position A resulted in a displacement of 0.6 μm at position A and 60 μm at output position B, indicating a magnification of 100.

However, 5 N appeared too large for a mere 0.6 μm displacement. Further analysis found that even larger forces are needed to produce larger displacement while the magnification ratio decreases with increasing displacement. To obtain a displacement of 1 μm needs 10 N and the magnification ratio shrinks to 68. To obtain 10 μm displacement will need over 100 N of force. This is undesirable. Input forces could be reduced by reducing the thicknesses of the flexure sections but then unfavorable deformation due to compression would increase. It appeared that this dilemma of using compressive monolithic hinges was difficult to resolve.

Therefore, it was decided to move to conceptual design D to use flat springs for the hinges.

### 2.5. Conceptual Design D: Assembled Structure of Multi-Planar Layout with Flat-Spring Couplers and Hinges

A.System layout

Through the studies of the previous conceptual designs, it was concluded that a better approach for the hinge would be flat-spring in tension, which could minimize unfavorable deformation as well as hinge sizes.

Figure 9 depicts the conceptual design of a single lever structure, which includes a base frame LB2, an arm (beam) LA2 and a hinge LH2 that connects the arm to the base frame. The flat-spring foil hinge is attached to a base mounting face P1L2 on the base frame and to one end of the arm. The amounting can be by screw or by joining such as brazing or soldering or glue. A small bare section of the foil spring (Ls_2_) is to facilitate the hinge function. The arm includes a protruded feature with a second mounting face P2L2, which is parallel to the first mounting surface P1L2. When the protruded feature on the arm is pushed downward (i.e., toward −z direction), downward displacement at the second mounting face P2L2 is amplified at the end of the arm at a third mounting face P3L2 by a factor of L_2_/*l*_2_, the ratio of the distances from the third mounting face P3L2 and the second mounting face P2L2 to the base mounting face P1L2.

Figure 10 explains the assembly of a magnification mechanism of three stages of the lever structures in exploded view. The three lever structures are of basically similar configuration as the one depicted in Figure 9. The lever structures are disposed side by side with all arms in parallel and with the plane of motion of each arm different but parallel to each other. The first stage lever L1 includes a base frame LB1, an arm LA1 and a hinge LH1. The arm LA1 also has a protruded feature as the contact place CP, onto which a ball, for example, made of Zircon, is mounted as the contact artifact 5 as the input point. The magnification ratio from the contact place CP to the end of the arm P3L1 is *L*_1_/*l*_1_. Lever *L*_1_ is oriented with the output end P3L1 of the arm LA1 pointing to the −x direction.

The second stage lever, L2, already described in Figure 9, is placed in parallel relative to lever L1, but oriented with its output end of arm P3L2 pointing toward the +x direction. That is, in order to have a compact form factor and small characteristic size, adjacent lever structures are aligned with the output ends of the arms pointing to opposite directions. Further, the second mounting face P2L2 of the arm LA2 is aligned to the end face P3L1 of the arm LA1 such that an L-shaped foil flexural foil, as described in Section 2.3, CF12 couples lever L1 and lever L2 at the two faces. The lower portion of the foil is mounted to the arm LA1 at its end face P3L1 and part of the upper portion is mounted to the second mounting face P2L2 of the lever L2. Thus, the displacement at the output end of the arm LA1 can be transmitted to the input end of the arm LA2 of lever L2.

The construction of the third stage lever, L3, and its coupling with the second stage are basically similar. The output end of the third stage lever is the measurement point MP.

The 3 lever structures are assembled side by side together, with a spacer stripe (SP12 and SP23) between adjacent lever structures to prevent unwanted sliding contact between the arms, and become an integral structure, as depicted in Figure 11. A tiny displacement at the contact place CP is magnified at the measurement place MP by a cascaded ratio of
Total theoretical magnification = (*L*_1_/*l*_1_) (*L*_2_/*l*_2_) (*L*_3_/*l*_3_) (1)

Arrows 10 and 20 indicate directions of the input micro-displacement and of the output magnified displacement of the assembly, respectively. Assuming the magnification of each lever structure is five, the total theoretical magnification is then 125, two orders of magnitude amplification.

B.Design of contact force and the flat spring hinges

A simplified analytical model was developed for an initial approximation design of the flat-spring hinges. The model assumes that, in an assembled stages of cascaded lever structure, output displacement and force can be transferred from one stage to the next stage by “perfect” couplings with no loss and all forces in the vertical direction are balanced. It is also assumed that, as a rough approximation, the bending of the hinge flat spring can be viewed as a simple cantilever beam. Accordingly, this simplified cascaded lever model can be analyzed in mechanics as three free bodies with each body supported by a small elastic simple beam, as illustrated in Figure 12.

By force balancing, relations among input force *F*_1_, inter-lever forces *F*_2_ and *F*_3_, lever arm weights, *w*_1_, *w*_2_ and *w*_3_, and geometric parameters can be formulated as follows:(2)$$F_{1}=\frac{\;M_{1}-w_{1}\cdot \frac{L_{1}}{2}+F_{2}\cdot L_{1}}{l_{1}}$$
(3)$$F_{2}=\frac{\;M_{2}-w_{2}\cdot \frac{L_{2}}{2}+F_{3}\cdot L_{2}}{l_{2}}$$
(4)$$F_{3}=\frac{\;M_{3}-w_{3}\cdot \frac{L_{3}}{2}\;}{l_{3}}$$
where the bending moments of the hinges can be approximated by the simple beam relation:(5)$$M_{3}=\frac{\theta_{3}\mathrm{EI}}{Ls_{3}}$$
(6)$$M_{2}=\frac{\theta_{2}\mathrm{EI}}{Ls_{2}}$$
(7)$$M_{1}=\frac{\theta_{1}\mathrm{EI}}{Ls_{1}}$$
with *θ*_3_, *θ*_2_ and *θ*_1_ as the angles of the rotation of the lever arms, *Ls*_3_, *Ls*_2_ and *Ls*_1_ are the sizes of bare sections of the hinge springs, E is the Young’s modulus of the material of the hinge spring, and area moment of inertia of the cross-section of the hinge spring expressed in terms of its thickness *t* and width *b* as:(8)$$I=\frac{bt^{3}}{12}$$

The input displacement, corresponding to the acting position of input force *F*_1_, is:(9)$$d_{in}=d_{1}=l_{1}\theta_{1}$$

Additionally, the displacements at position 2 and 3 are:(10)$$d_{2}=l_{2}\theta_{2}=\frac{L_{1}}{l_{1}}d_{1}$$
(11)$$d_{3}=l_{3}\theta_{3}=\frac{L_{2}}{l_{2}}d_{2}=\frac{L_{2}}{l_{2}}\frac{L_{1}}{l_{1}}d_{in}$$

Rearranging Equations (3)–(11) and inserting them into Equation (2) gives the expression of input force *F*_1_ as a function of input displacement *d_in_* and system parameters:(12)$$F_{1}=\left[\frac{d_{in}\mathrm{EI}}{Ls_{1}l_{1}}-\frac{w_{1}L_{1}}{2}\right]\frac{1}{l_{1}}+\left[\frac{d_{in}L_{1}\mathrm{EI}}{Ls_{2}l_{1}l_{2}}-\frac{w_{2}L_{2}}{2}+\left(\frac{d_{in}L_{1}L_{2}\mathrm{EI}}{Ls_{3}l_{1}l_{2}l_{3}}-\frac{w_{3}L_{3}}{2}\right)\cdot \frac{L_{2}}{l_{3}}\right]\cdot \frac{L_{1}}{l_{1}l_{2}}$$

By applying Equation (12), the dimension of the hinge flat-spring can be designed to match the suitable range of contact force. Conceptual design D uses aluminum as the lever structures, spring steel foils as hinges and takes key parameters set or learned from previous designs:*l*_1_ = *l*_2_ = *l*_3_ = 10 mm 
*L*_1_*= L*_2_*= L*_3_ = 50 mm 

Lever arm cross-section 8 mm × 8 mm:*Ls*_1_ = *Ls*_2_ = *Ls*_3_ = 1 mm.

With these material and geometric information, the relations between input force and input displacement at different hinge dimensions were computed and shown in Figure 13 and Figure 14. By selecting flat-springs of thickness *t* = 0.1 mm and width *b* = 8 mm, the input contact force will be from zero to slightly over 8 N in the full 10 μm range of the measurement. This suggests that this design appeared viable.

## 3. Prototypes and Tests

Prototypes were developed in three phases. In the first phase a model of three stages of lever of comparatively large size, 82 mm in length, was built and tested to verify the basic design concept and metrology parameters. In the second phase, compact models of 25 mm long with two stages of levers were built and tested. Finally, the compact model was modified and packaged into an Alpha model with features of a pre-commercial model.

### 3.1. 82 mm Proof of Concept Prototype

A first prototype built was a proof-of-concept prototype using the design parameters concluded in Section 2.5 B. Figure 15 shows a photograph of the prototype displacement magnifying mechanism. This 82 mm long mechanism looks basically identical to the depiction of Figure 11, except that flat-spring hinges and couplers were mounted by screws and washers. The lever arms and bases were made from aluminum by wire EDM. A stainless-steel ball was used as the contact ball. A first surface reflector was attached to the upper surface of the output end B, to be used for optical lever measurement. A hole was opened on the base frame underneath the output end B for mounting sensors.

In order to test and calibrate the prototype, a testing/calibrating setup was built. As shown in Figure 16, the setup was basically a big lever with a metal push beam supported at the left end as fulcrum (F) and at right end by a vertically erected linear stage with a digital micrometer head as the effort end (X). The prototype displacement magnifying mechanism was placed close to the fulcrum underneath the push beam with the contact ball in contact with the lower surface of the push beam at a load point (Y). The lever ratio of FX to FY was roughly 10:1. Thus, when the micrometer head pushed up or lowered down the push beam at the effort end by 1 μm, an input displacement of about 0.1 μm to the prototype could be generated. An eddy current sensor (sensor 1), Karman KD2306 1U1, was placed to the side of the contact ball of the prototype to precisely measure the input displacement at load point Y as the reference for the calibration.

Three different methods were applied to test/calibrate the prototype displacement magnifying mechanism, with the aim to verify the feasibility of the design, and most importantly, to test the repeatability of the hardware implementation.

A.Measuring prototype output displacement by a second eddy current sensor

A second eddy current sensor (sensor 2 in Figure 16), Micro-Epsilon eddyNCDT 3100 EPS08, was placed at the output sensor mounting seat under the output end B of the prototype mechanism (Figure 15) to measure output displacement and to check the magnification ratio and measurement repeatability.

Initially, it was observed that the output displacement at B measured by the second eddy current senor exhibited vibrations with amplitudes of 40–50 μm even when the input end at A was stationary. It was suspected that the nearly overhanging arm of the L3 lever was picking up the floor vibrations caused by frequent movements of members of our laboratory, which is on the fifth floor of the engineering building. A simple damper with a small acrylic piston in a small open oil cup was added underneath the lever arm (D, in Figure 15) and basically eliminated the excess vibration.

Three measurements were performed to correlate the input displacement *d_in_*, as measured by the first eddy current sensor under the push beam of the calibration setup, to the output displacement *d_out_*, measured by the second eddy current sensor. In each measurement, the input was set relative to an arbitrary starting position and changed by adjusting the micrometer head of the test setup to lower down the push beam and thereby pushed down the contact ball. The output was also recorded as change relative to a starting position corresponding to the staring position of the measurement. For repeated measurements, the test setup and the prototype were returned to the original starting position as measured by the readout signal of the first eddy current sensor, to ensure that all measured curves have the same origin.

The results are shown in Figure 17. As the second eddy current sensor used for measuring the prototype output has a maximal range of only 0.8 mm, the calibration could only cover 6 μm of the input displacement. The slopes of the measured curves indicate that the magnification ratio exceeds 150:1 in the first 2 μm and then approaches 100:1. The unidirectional repeatability R_i_ of the prototype were estimated according to ISO 230-2 [18] by applying the following relations:(13)$$s_{i}↑=\sqrt{\frac{1}{n-1}\sum_{j=1}^{n}{\left(x_{ij}↑-\bar{x}↑\right)}^{2}}$$
(14)$$R_{i}↑=4s_{i}↑\;$$

Take note that, although the “actual measurement” was performed on the magnified displacements, the “effective measurement” of the prototype device was for the pre-magnified input micro-displacement. In applications, measured output displacement will be converted into input micro-displacement using the measured curves. Therefore, the quadruple standard deviation 4*s_i_* was estimated from the output displacements, but the effective repeatability should be mapped to the input displacement according to the measured curves, as highlighted in Figure 17. Thus, the repeatability of the prototype mechanism, based on the measurement by the second eddy current sensor, was estimated to be 0.005–0.2 μm over the 6 μm range, which is reasonably good, even considering the small range around 5.5 μm that gave the 0.2 μm value.

B.Measuring prototype output displacement by an optical lever

As the above measurement by the second eddy current sensor was only able to cover a 6 μm range, a laser optical lever method was further conducted to verify the measurement repeatability of the prototype mechanism covering the targeted full 10 μm measurement range.

Figure 18 illustrates the setup of the laser optical lever. A laser was mounted above the testing/calibrating setup with the laser beam pointing downward. A wide notch was opened on the push beam so that, when the prototype mechanism was positioned underneath the push beam for calibration, the L3 lever arm and the first surface reflector on it (Figure 15) was exposed to the laser above. The laser beam hit the first surface reflector, reflected upward, hit another reflector fixed above the test setup and then was redirected to a distant screen of graph paper, as shown in a photograph of the laser spot on the screen.

Three measurements were conducted following a procedure similar to the one described in the previous sub section, except that the output values were distances of movement of the laser spot on the screen. Figure 19 shows the results of measurements. The unidirectional repeatability R_i_ over the 10 μm measurement range was estimated, by the similar method used in Figure 17, to be 0.05–0.2 μm, which is in good consistency with the results from the eddy current sensor measurement. The accuracy A of the prototype mechanism was also estimated, according to the same ISO 230-2 standard by applying the following relation:(15)$$A=\max\left[{\bar{x}}_{i}+2s_{i}\right]-\min\left[{\bar{x}}_{i}-2s_{i}\right]\;$$
and was calculated to be 0.4 μm.

C.Measuring prototype output displacement by a Hall sensor

A Honeywell SS495A Hall effect sensor was mounted to the output sensor mounting seat of the 82 mm prototype mechanism (Figure 15). A tape magnet, cut from magnetic tape obtained from the Misumi Corporation, was fixed to the lower surface of the output end facing the Hall sensor (C, in Figure 15). An Arduino Uno board was used to read the output voltage from the Hall sensor and send it to a laptop PC for records. A prototype sensor device was thus constructed.

Before testing the prototype sensor device, a separate test on the Hall sensor was conducted by moving a tape magnet relative to the sensor. The output voltage to input displacement relation shows very good linearity and consistency, as shown in Figure 20. The Arduino Uno had an analog to digital conversion resolution of 10 bit and the Hall sensor had a voltage output scale of 5 V. Therefore, the resolution of voltage measurement was 0.005 V (0.1%). The standard deviation of the measurements was 0.006 V, which was likely to reflect the resolution of the voltage measurement of the Arduino Uno. 0.006 V maps to a repeatability of 12.5 μm.

However, it should be noted that a separate calibration, like in Figure 20, cannot be applied to calibrate the prototype displacement magnifying mechanism that used a different piece of tape magnet in a slightly different geometric environment, because it was found that neither the magnetic polarity nor the strength is uniform over the surface of the commercial magnet tape. Therefore, there is almost no way to obtain two pieces of small tape magnets with the same magnetic properties.

In testing the sensor device prototype, again, three measurements were conducted following a procedure similar to the one described in sub section A, except that the output values were the output voltage of the Hall sensor. Figure 21 shows the results of the measurements. Again, the quadruple standard deviation was estimated from the Hall sensor output voltage, but the effective repeatability and accuracy mapped to the input displacement according to the measured curves, as highlighted in Figure 21. Thus, the repeatability of the prototype mechanism was estimated to be 0–0.4 μm over the 10 μm range and the accuracy was 0.7 μm.

To sum up in other words, when applying the prototype sensor device to measure a micro-displacement, a best-fit curve will first be generated from the curves in Figure 21. Then a measured voltage output will be converted to a micro-displacement according to this best-fit curve. The converted micro-displacement will have a repeatability between 0 and 0.4 μm and an accuracy of 0.7 μm.

Thus, the feasibility of the design and the ability of the prototype device to measure micron level displacements at sub-micron repeatability and accuracy are verified and confirmed.

### 3.2. 25 mm Compact Model

Following the design procedure and based on the design features of the 82 mm proof of concept prototype, a compact 25 mm model was designed and developed. The compact model also has a measurement range of 10 μm and repeatability and accuracy in the sub-micron level but uses only two lever structures.

Figure 22 shows the final design and key dimensions of the compact model. The magnification ratio was designed to be about 127. The thickness and feature dimensions of the flexural foils were determined from finite element simulations with three major goals. The first design goal was to keep the contact force below 0.3 N, close to commercial contact type displacement probes. The second goal was to make sure the flexural foils to work within endurance limit. Flat spring foils were strip steel products from Sandvik [19]. Analysis and finite element simulation of the model were conducted to check that stresses in the hinges and couplers were way below the endurance limit of the material. Additionally, the third goal was to maintain the magnification ratio of the mechanism under different model orientations, that is, when the direction of gravity acting on the model is different. Simulation showed a magnification ratio close to 125 and direction of gravity has a negligible effect on it.

The aluminum bodies of the lever structures were made by wire EDM and CNC milling. The flexural foils were cut by laser. Due to the small size of the model, a set of fixtures were made and a step-by-step procedure was developed to ease assembly. Figure 23 is a CAD model depiction of the use of the fixtures in the assembly of the model. Figure 24 shows a picture of the compact model with a Hall sensor and a tape magnet mounted.

The relationship between input force and input displacement of the compact model was measured by placing the compact model on a precision scale and applying a vertical micro-stage to push on the contact ball of the compact model. Figure 25 shows the measured results on three different compact models, but all made to the same specifications. The input force is below 0.15 N within the 10 μm range.

Tests and calibrations of the compact model were conducted using the setup of Figure 16 following the same procedure and the same electronic readout setup used on the 82 mm prototype with the Hall sensor. Figure 26 shows the compact model set in measurement position in the setup. It was found that the floor vibration effect on the output lever, encountered in the initial tests of the 82 mm prototype, was not seen in the compact model.

Figure 27 shows the results of the measurements. Note that the trend of decreasing voltage w.r.t. increasing displacement in Figure 27 is different from Figure 21, because the tape magnets used in the two cases had different polarities. The repeatability of the compact model was estimated to be 0–0.7 μm over the 10 μm range, with the poorest number occurring in a narrow range around 3 μm displacement. The accuracy was 0.5 μm. As for the measurement resolution, the maximal voltage difference over the full measurement range was 0.53 V, while the Arduino Uno readout board has a resolution of 10 bits over a full voltage range of 5 V, which corresponds to a voltage resolution of 0.005 V, which is about 1/1000 of 0.53 V, thereby the displacement resolution is also about 1/1000 or 0.1 μm.

### 3.3. LCMDS Alpha Model

Based on the 25 mm compact model specifications, an Alpha Model of the LCMDS was designed and developed. The Alpha Model design includes a few modifications over and additions to the compact model in order to be as close to a commercial model as possible.

First, a package is added to the MDMM compact model. The package basically includes an aluminum base, to which the MDMM is mounted, and an acrylic protective casing, consisting of a cover screwed to walls, as shown in a side view of the design in Figure 28.

A special Contact Relay Mechanism (CRM) is attached onto the inside surface of the cover of the package casing. The purpose of the CRM is to avoid direct contact between the MDMM contact ball and an external target surface to be measured, thereby protecting the internal mechanism from any possible disturbance or contamination. The CRM includes a thin but wide flexural cantilever beam with its fixed end attached to the cover and its free end attached with a seat holding a contact ball. A small opening on the cover allows the CRM contact ball to stick out above the top surface of the cover. The CRM contact ball is the ball to be in actual contact with a surface to be measured. The cantilever beam was made from 0.1 μm thick spring steel and had certain stiffness to limit the displacement of the CRM ball and its seat to be only in a direction normal to the cover. The lower surface of the ball seat was made to be in contact with the contact ball of the MDMM with some preload force. Therefore, any displacement is transferred fully to the internal MDMM. Figure 29 depicts the construction of the CRM.

In addition, a pre-screening procedure for magnet tape material was developed and conducted to select areas of strong magnetic field to be used with the Hall sensor. The procedure basically scans a Hall sensor over surface of a magnet tape and records voltage output values to create a magnetic strength map of a tape. As a result, the maximal difference of voltage output over the full measurement range was significantly increased.

Further, the foil thickness of the flexural hinge of the second lever was increased from 0.01 mm to 0.02 mm in order to increase the stiffness of this output lever for improved anti-vibration capability. The modification increases the first mode natural frequency from 33 Hz to 102 Hz, above the 10–55 Hz range used in some commercial contact type displacement sensors, for example, the Keyence digital contact sensor [5].

Further, the MDMM design used in the compact model was modified with a mechanically balanced lever 2 arm. As can be seen from the picture in Figure 24, the L2 lever arm slacks slightly downward, because it is supported basically at one end by its hinge and coupler with its center of mass overhanging. As a result, if the sensor device is to be used in an orientation different from the erected orientation, such as in horizontal orientation for measuring sideway displacements, the magnet-to-sensor gap size at zero-input force state will be different (larger than that of the erected orientation, since gravity component is smaller). However, this does not prevent the sensor from functioning, the difference between the maximal and the minimal of output voltage over the full measurement range will become smaller, thereby reducing sensor resolution and sensibility. Figure 30 shows the design to adjust the location of the center of mass of the L2 lever arm. Basically, some mass was removed from the lever arm and an extension with a brass counterweight was added to the opposite end of the arm. Ideally, the center of mass should be located in the middle of the bare section of the hinge, so that arm mass poses no moment to the rotation center of the arm. However, allowable space under the limitations by the casing wall and hinge clamp on the lever base did not permit the ideal design. A compromise was made to lower the center of mass by about 1.2 mm, as indicated by “adjusted center of mass” in Figure 30.

Figure 31 shows a picture of the completed Alpha Model in side view. Figure 32 shows two packaged devices, with the Alpha Model in the foreground.

Tests and calibrations of the Alpha Model were conducted using a second calibration setup, which is capable of positioning and testing a packaged or unpackaged prototype in both erected and horizontal orientations. Figure 33 shows the new setup operating in horizontal orientation. Similar to the setup of Figure 16, a manual linear micro-stage actuates the push beam at location X. The load point Y sees 1/5× of the displacement of the micro-stage, which was precisely measured by a Keyence CL-L030 color confocal laser displacement sensor. The Keyence sensor has an accuracy of ~1 μm and a resolution of ~0.25 μm. Therefore, at load point Y, setting a displacement of 0.2 μm accuracy and 0.05 μm resolution is achiveable.

The results of calibration are shown in Figure 34. The curves of the two orientations appeared quite similar and the maximal difference of voltage output over the full 10 μm range was increased to over 1.4 V in both cases. Uni-directional repeatability and accuracy, estimated from 3 measurements by the same method used previously, and resolution, based on using the 10-bit Arduino Uno readout board, are summarized in Table 3.

## 4. Discussions of Related Issues

### 4.1. Effect of Variations of Part Dimensions and Assembly Precisions

Assembly of parts inevitably introduces relative positioning errors among parts. In practice and for cost reason, no two MDMMs can be manufactured and assembled to identical dimensions. As tape magnets cannot be identical either, as already mentioned, future commercial LCMDS devices will need calibration individually before shipping. Therefore, it will be convenient if positioning variations in assembly can be tolerated to a certain extent as long as the magnification ratio does not change significantly and measurement repeatability is preserved in each device.

Test results support such tolerance for assembly variations. Parts for at least three prototypes of the 25 mm MDMM compact model were fabricated, assembled and tested during the development process and all three prototypes worked with similar repeatability and accuracy in a measurement range of 10 μm. It is reasonable to assume that small variations of part dimensions and assembly precisions existed among these prototypes. However, these small variations did not appear to affect the functionality of the prototypes, as shown by the test results of two of the prototypes from Figure 27 and Figure 34.

### 4.2. Effect of Temperature

Thermal expansion and contraction will cause a change of length of a lever arm. Therefore, the question is: Does temperature variation cause change of magnification ratio? From Equation (1), we can see that when temperature varies the lengths of load (L_1_~L_3_) and the lengths of effort (*l*_1_~*l*_3_) will expand, or contract, by the same proportion. Therefore, the expansions or contractions will cancel each other out and a constant magnification ratio will be maintained.

### 4.3. Effect of Magnetic Disturbance

The use of Hall sensor and magnet makes the LCMDS device potentially sensitive to external magnetic disturbances. To understand how significant the effect can be, variations of output voltage of the Alpha Model were measured with a neodymium magnet placed at different distances from and in different orientations with respect to the Alpha Model. It was found that at a distance of ~40 mm the neodymium magnet began to affect the voltage output, and the corresponding magnetic field strength experienced by the Alphas Model at that distance was measured to be about 0.2 mT (or about three to eight times the magnitude of the geomagnetic field) at the Alpha Model. That is, unless an external magnetic disturbance higher than three to eight times that of the earth is imposed on the Alpha Model, the device will not be affected.

### 4.4. Design for Low-Cost Small Lot Size Production

With the performance of the Alpha Model demonstrated and related issues addressed, current planning of the next step of development is to enable low-cost small lot size production of a Beta Model with size and features comparable to the Alpha Model.

The BOM (bill of materials) cost of the Alpha Mode was slightly over US$350, within which machining costs were a significant portion. The design and manufacturing of the Beta Model will have two features to reduce costs. The first feature is that the MDMM was redesigned to reduce number of parts and to make all parts into extruded geometries that can be joined by glue rather than by screws. Figure 35 illustrates the idea of part shape simplification by extruded geometry. Figure 35a shows the design of the L1 lever arm of the Alpha Model, which requires precision machining to make a few small features. Figure 35b shows the simplified design, which, except for the circular recess at the top, is basically an extrusion of a 2D geometry.

The second feature is that each part will be made by a process using a multi-part intermediate object. As all parts are designed as extruded geometries, multiple parts of the same geometry can be included into one multi-part intermediate object, which is a single workpiece of the same cross-sectional geometry as the part and this multi-part intermediate object can be produced by machining at a cost comparable to a small single part but effectively at a much lower per part cost. For example, Figure 35c shows a 4-part intermediate object including 4 pieces of the lever arm of Figure 35b. After the multi-part intermediate object is machined, individual parts can be separated by wire EDM, which will be a one-time operation of repetitive cut at a low cost.

Figure 36a shows an exploded view of the Alpha Model (excluding the casing and the CRM), which included 20 parts and 12 screws. After simplification, the designs and arrangement of parts are almost completely changed as shown in Figure 36b, which includes 12 parts (including a side wall that is part of the casing) and 4 screws. By using an eight-part intermediate object, the BOM cost of one Beta Model was estimated, based on quotations, to be less than US$100.

## 5. Conclusions

A new contact type compact displacement sensor device of sub-micron accuracy with low-cost potential was developed and tested. The basic concept combines a mechanical magnifying mechanism of side-by-side placement of lever structures with flexural hinges in a multi-planar layout with a Hall sensor to achieve good repeatability, compactness and low-cost. Prototypes of two different sizes and constructions¸ including an 82 mm long proof-of-concept prototype using three stages of lever structures and 25 mm long compact models using two stages of levers, with mechanical magnification ratios over 100 made and tested. A compact Alpha Model, which included a compact model mechanism with a balanced lever arm and a contact relay mechanism in a protective package, was shown to have a repeatability of 0.20–0.54 μm, an accuracy of 0.36–0.42 μm and a resolution of 0.03–0.04 μm in a measurement range of 10 μm. An important finding is the use of the L-shaped coupling foils for coupling levers placed side-by-side and the critical design parameters of the bare section gaps of the foil, which determine the effectiveness of transferring displacements and accommodating relative motions between the adjacent lever arms. To the authors’ knowledge, this is the first study and demonstration of using multiple stages of lever structures in side-by-side multi-planar layout for precise displacement magnification with a magnification ratio over 100. Part design modifications and corresponding manufacturing procedure for small lot size production were designed to provide an estimated bill of material cost per device under US$100. The low-cost devices can potentially be applied in quantities in measuring errors of machine stages in multiple directions, monitoring machine status and monitoring process status.

## 6. Patents

Key technical contents of the work reported in this paper are currently patent pending in Taiwan, Mainland China and the United States [20,21].

## Figures and Tables

**Figure 1 sensors-23-00326-f001:**
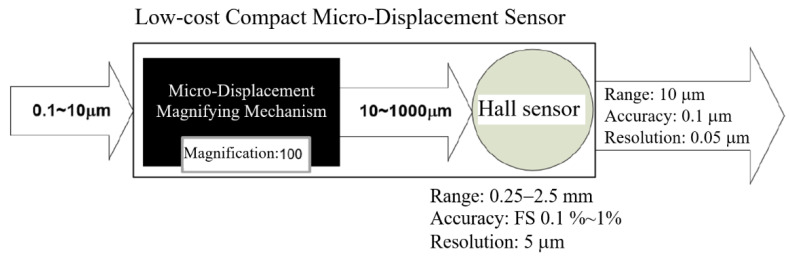
Basic concept of the Low-cost Compact Micro-Displacement Sensor (LCMDS) using a mechanical magnification ratio of 100 as an example.

**Figure 2 sensors-23-00326-f002:**
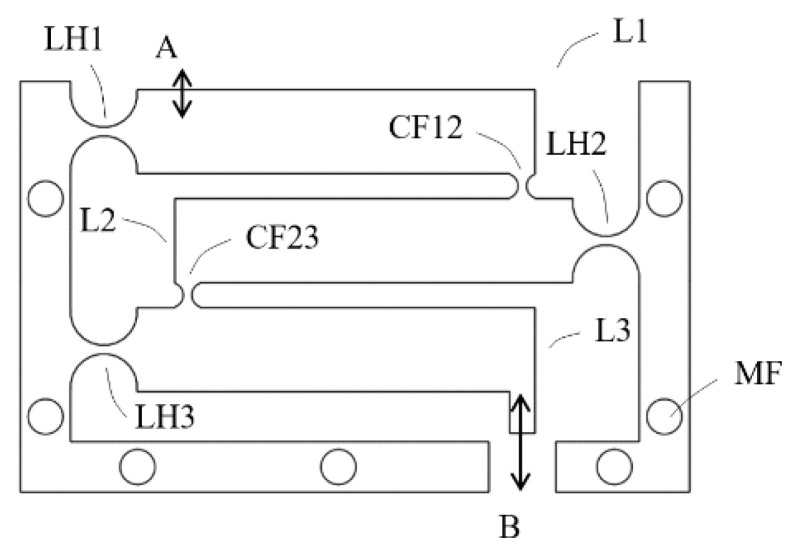
Conceptual design A, lever arm height 7 mm, width 5 mm, flexural hinge radius 3 mm, flexural section thickness 1 mm, coupling joint radius 1 mm, flexural section thickness 1 mm; mechanical advantage of each lever 5; MF indicating holes for mounting onto a back plate.

**Figure 3 sensors-23-00326-f003:**
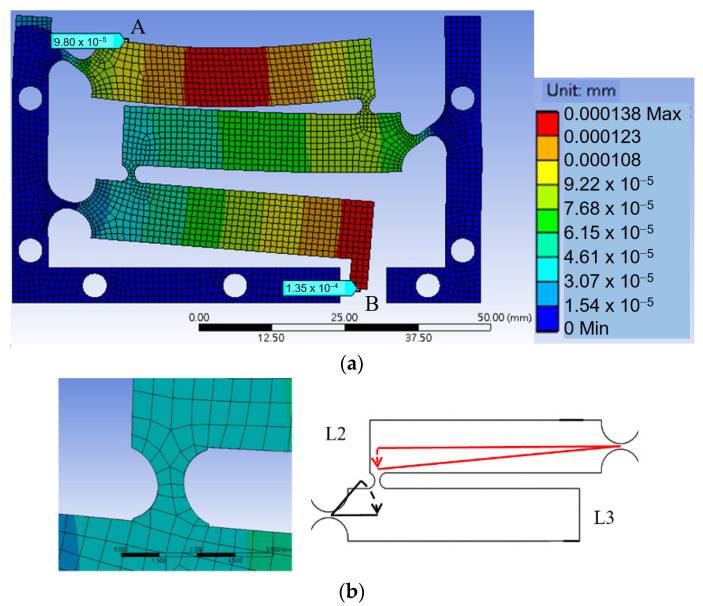
(**a**) is a total deformation graph of the structure when a force of 1 N is applied downward at contact place A; (**b**) is a close-up view of joint CF23 and exaggerated motion analysis between L2 and L3.

**Figure 4 sensors-23-00326-f004:**
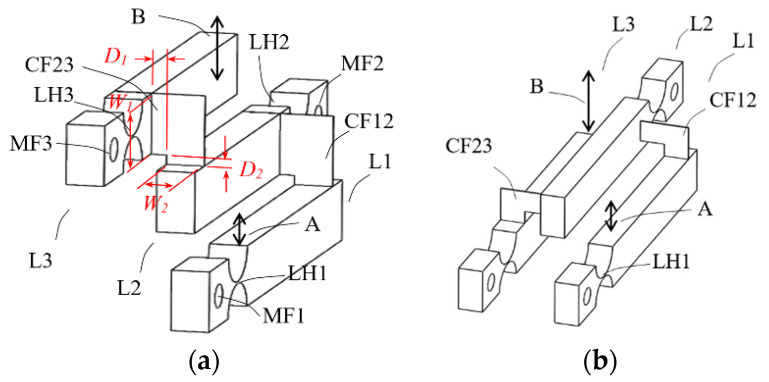
Conceptual design B, assembled structure of multi-planar layout with monolithic hinges, but “L-shape” flat spring couplers between levers; (**a**) coupler CF23 is “pull” type; (**b**) CF23 is “push” type.

**Figure 5 sensors-23-00326-f005:**
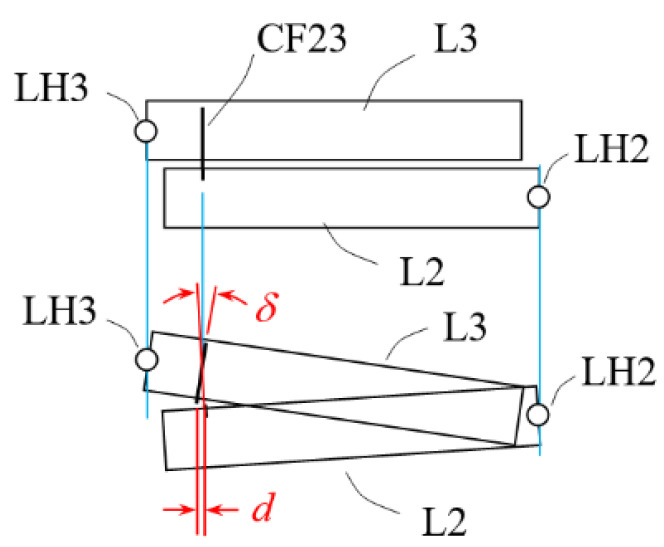
Exaggerated depiction of relative translational displacement *d* and rotational displacement *δ* between two adjacent levers at the location of the coupler, in the conceptual design B.

**Figure 6 sensors-23-00326-f006:**
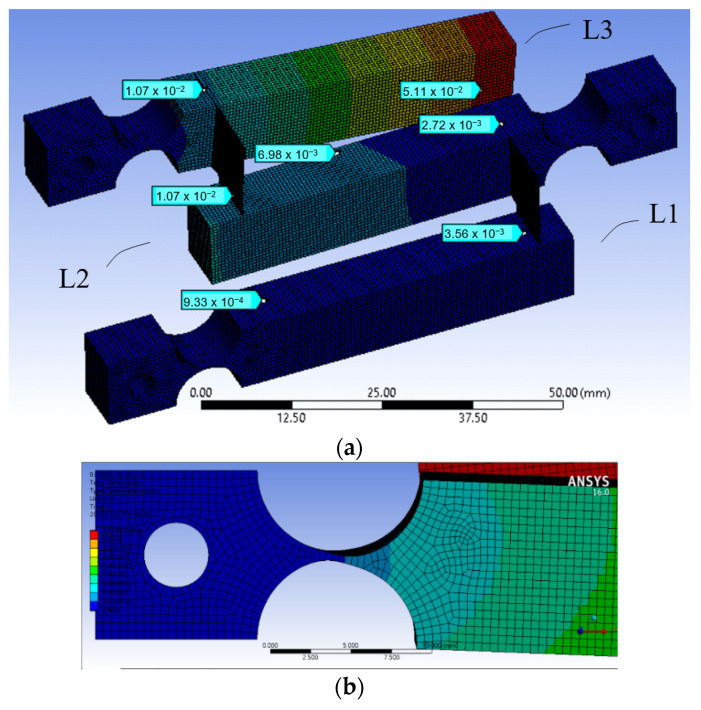
(**a**) A finite element model of the conceptual design B; (**b**) undesired shear deformation found in one of the monolithic hinges.

**Figure 7 sensors-23-00326-f007:**
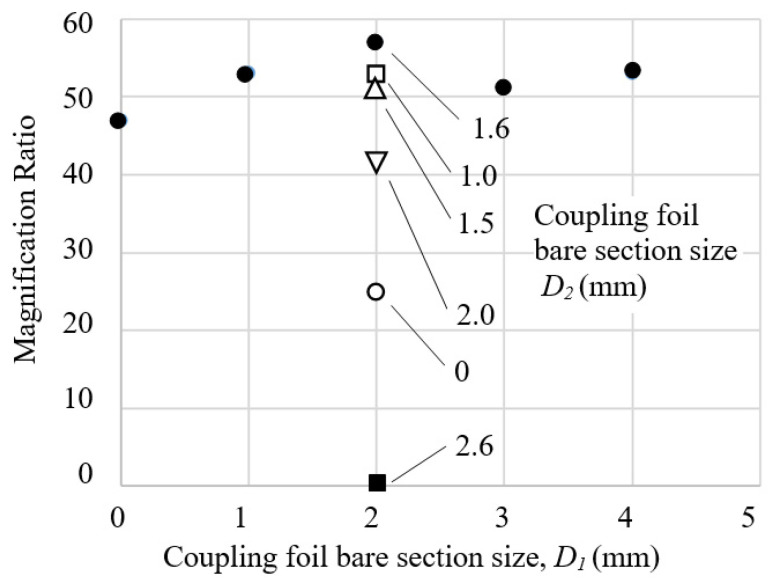
Effect of the sizes of bare sections of the coupling foils on the total magnification ratio of the conceptual design B of Figure 6a.

**Figure 8 sensors-23-00326-f008:**
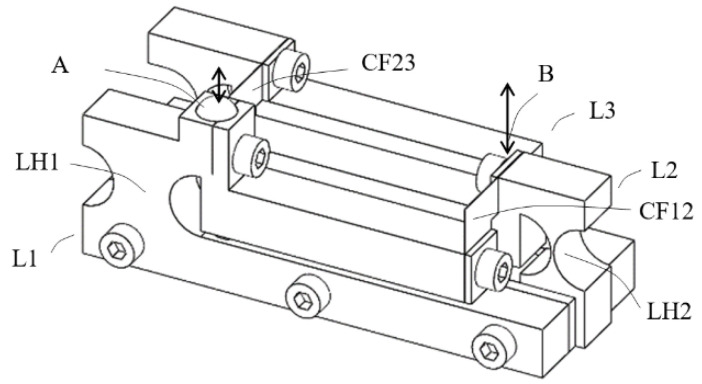
The conceptual Design C.

**Figure 9 sensors-23-00326-f009:**
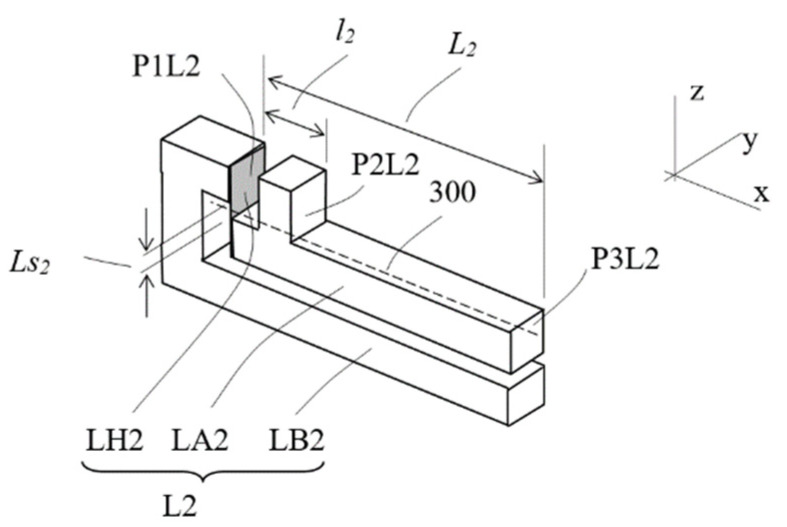
Single lever design of the conceptual design D.

**Figure 10 sensors-23-00326-f010:**
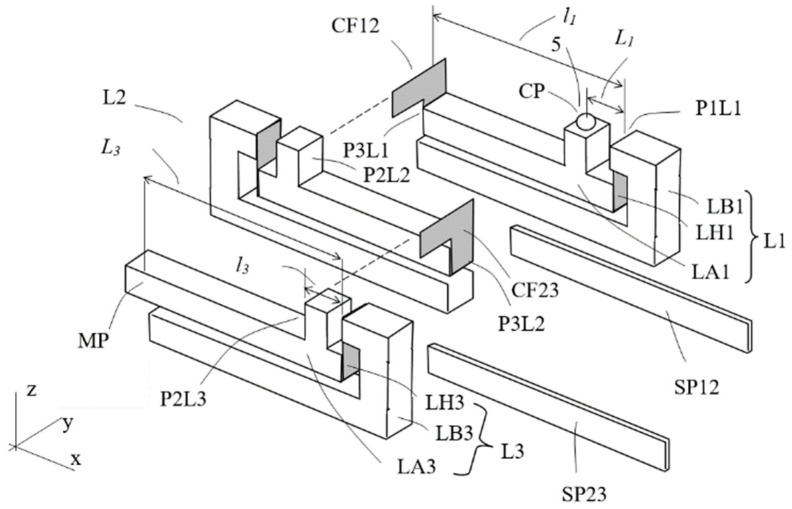
The conceptual design D in exploded view.

**Figure 11 sensors-23-00326-f011:**
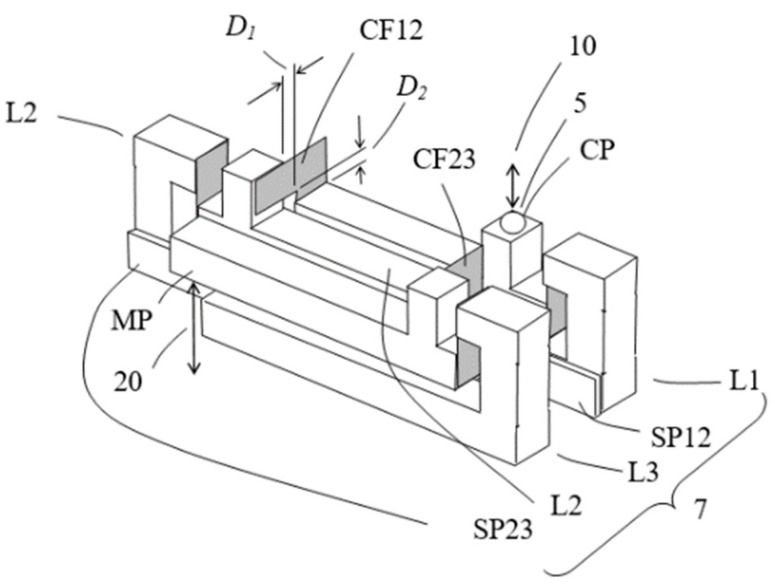
The MDMM conceptual design D.

**Figure 12 sensors-23-00326-f012:**
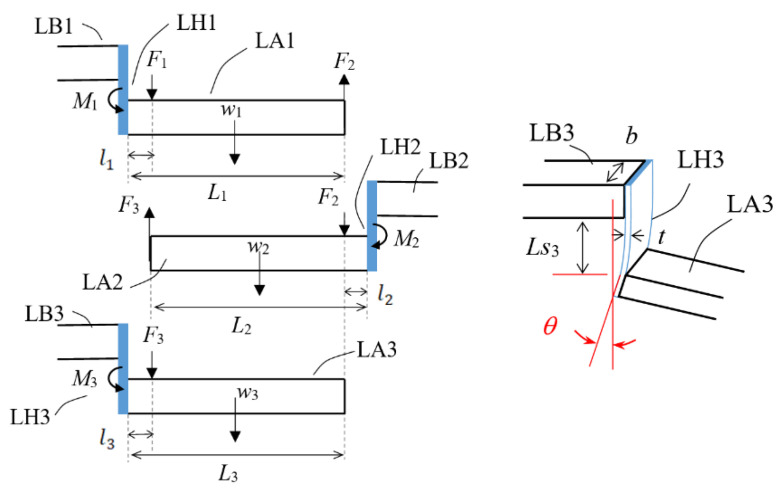
A simplified analytical model of the cascaded lever structures.

**Figure 13 sensors-23-00326-f013:**
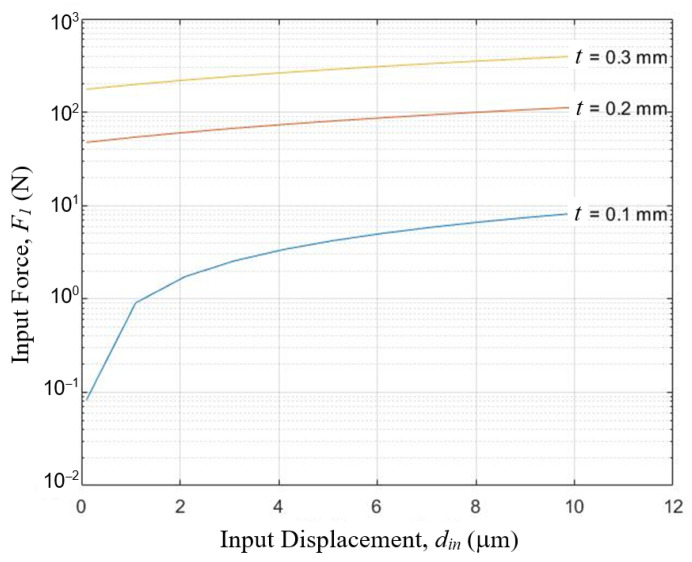
Relations between input force and input displacement of a three-stage cascaded lever structure at different hinge thickness *t*, when hinge width *b* = 8 mm, as computed by Equation (12) of the simplified analytical model.

**Figure 14 sensors-23-00326-f014:**
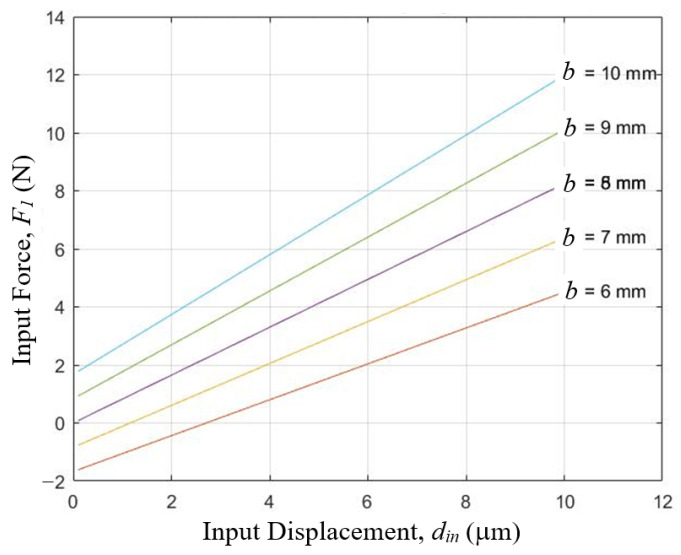
Relations between input force and input displacement of a three-stage cascaded lever structure at different hinge width *b*, when hinge thickness *t* = 0.1 mm, as computed by Equation (12) of the simplified analytical model.

**Figure 15 sensors-23-00326-f015:**
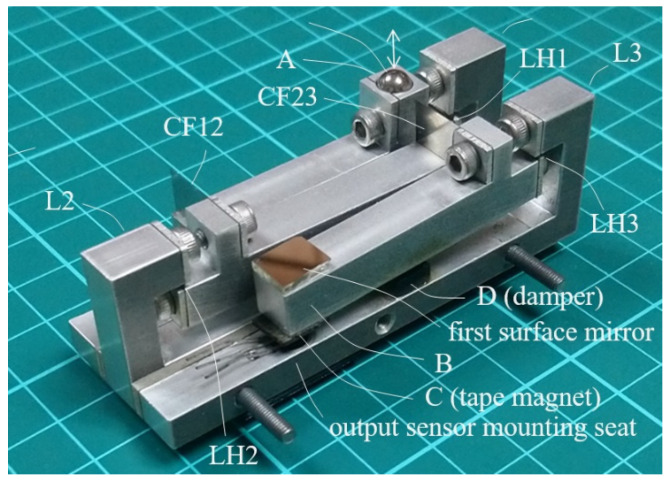
The proof-of-concept prototype, with an overall dimension of 82 mm × 28 mm × 24 mm.

**Figure 16 sensors-23-00326-f016:**
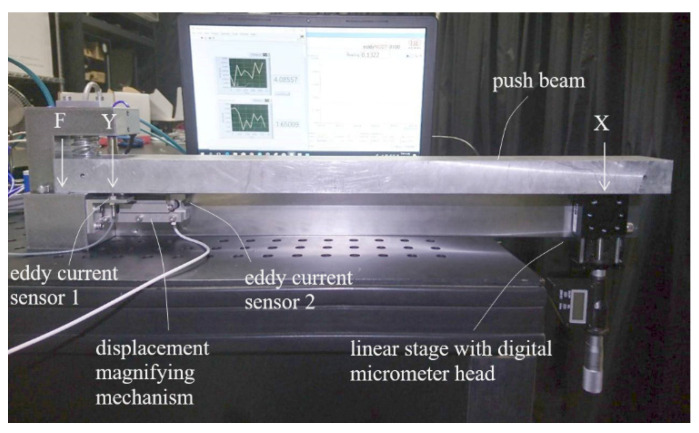
Setup for testing/calibrating the LCMDS prototypes.

**Figure 17 sensors-23-00326-f017:**
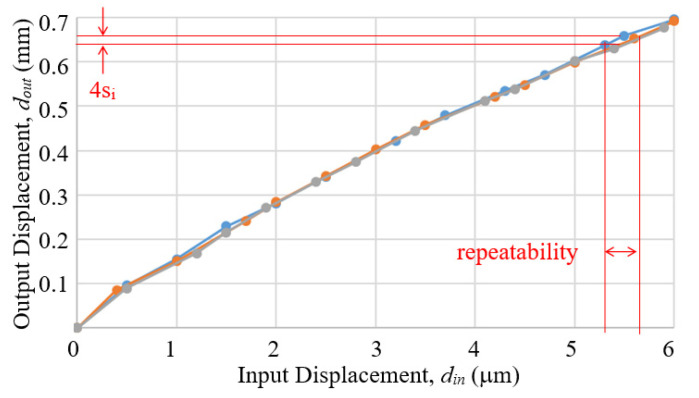
The results of the calbiration of the proof-of-concept prototype, by using an eddy current sensor to measure output displacements from three separate measurments.

**Figure 18 sensors-23-00326-f018:**
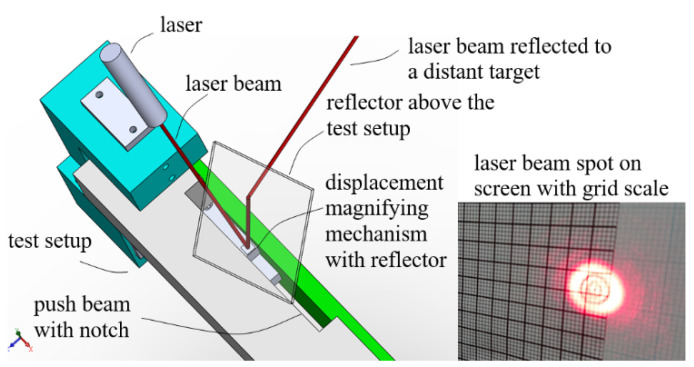
Optical lever setup illustration and a photograph of the laser spot on a distant screen.

**Figure 19 sensors-23-00326-f019:**
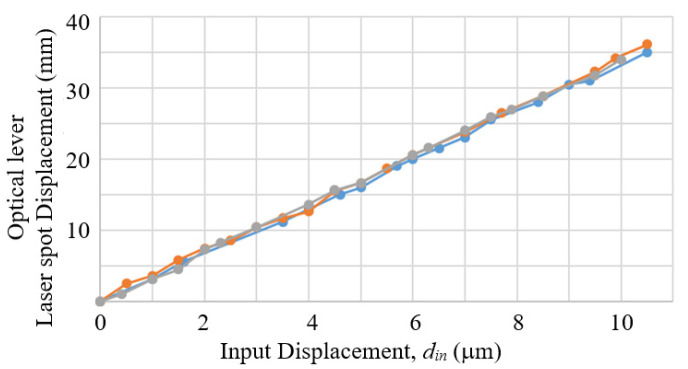
Results of three separate measurements of the proof-of-concept prototype by using the optical lever.

**Figure 20 sensors-23-00326-f020:**
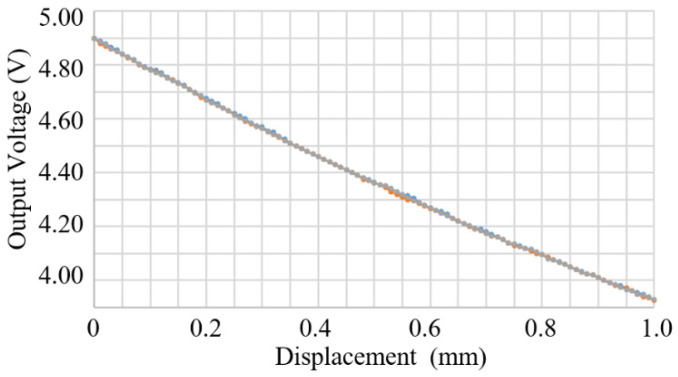
Output voltage of a Honeywell SS495A Hall senor in relation to the displacement of a tape magnet with respect to the sensor, three measurements relative to one arbitrarily set positon as the origin.

**Figure 21 sensors-23-00326-f021:**
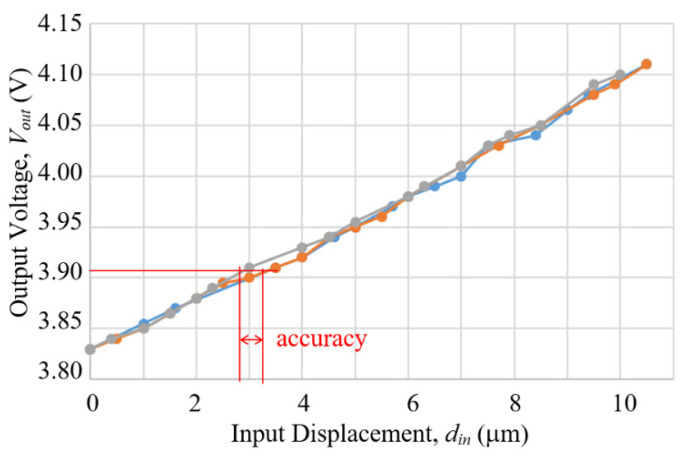
Results of three separate measurements of the proof-of-concept prototype by using a Hall sensor and a tape magent to measure output displacement, increasing input displacement meaning decreasing the gap distance between the Hall sensor and the tape magnet at position B on the third arm.

**Figure 22 sensors-23-00326-f022:**
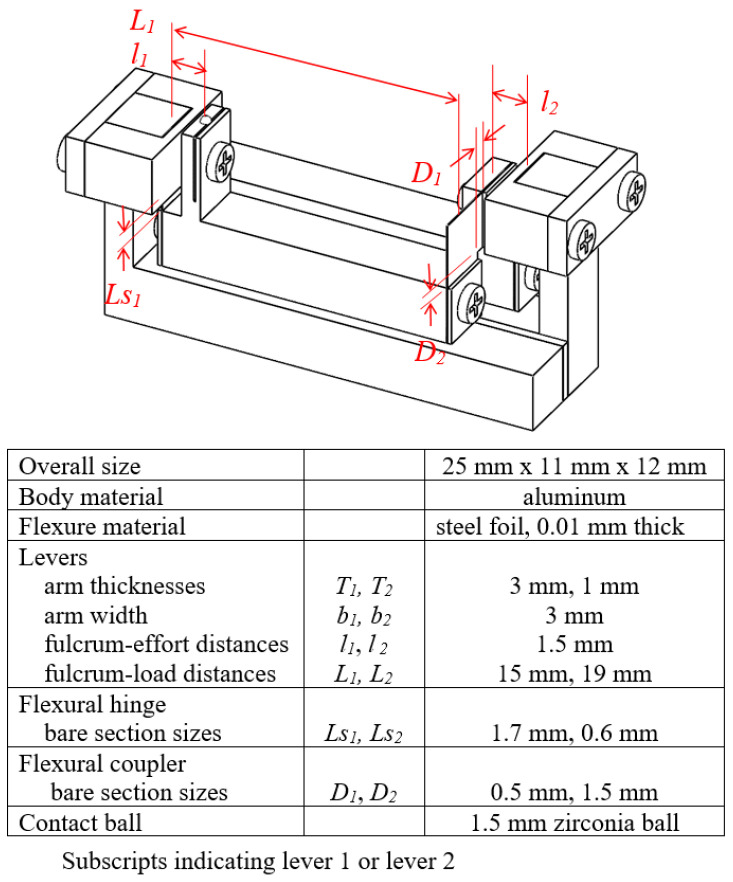
Layout and key dimensions of the MDMM compact model.

**Figure 23 sensors-23-00326-f023:**
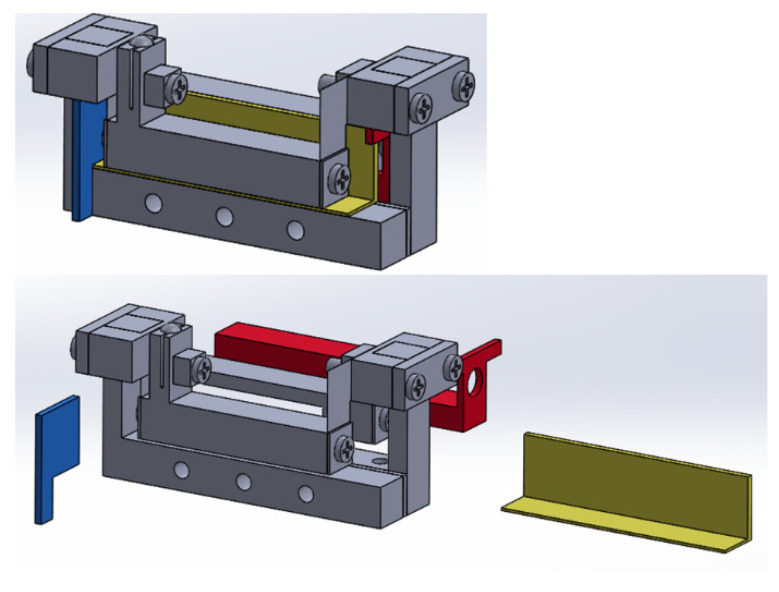
Fixtures for the assembly of the compact model.

**Figure 24 sensors-23-00326-f024:**
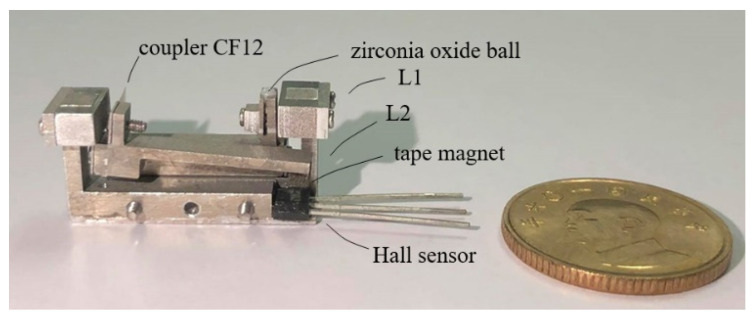
A prototype of the compact model, with an overall dimension of 25 mm × 11 mm × 12 mm.

**Figure 25 sensors-23-00326-f025:**
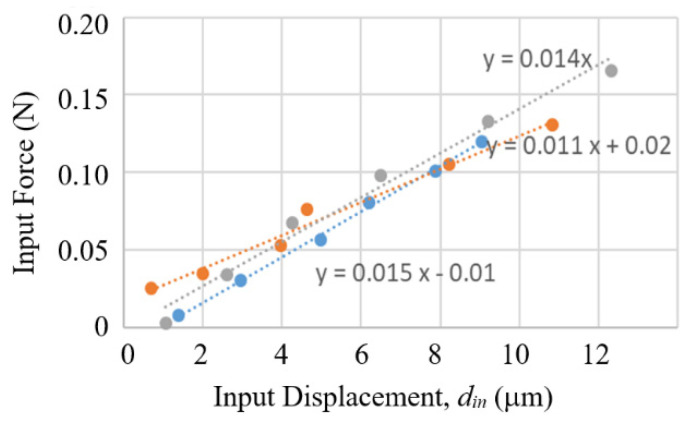
Input force in relation to input displacement of the compact model measured from three different prototypes of the compact model.

**Figure 26 sensors-23-00326-f026:**
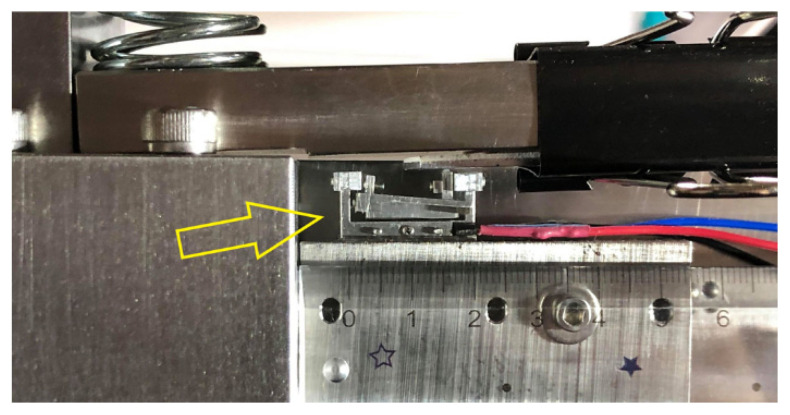
A prototype of the compact model in the measurement position in the setup of Figure 16.

**Figure 27 sensors-23-00326-f027:**
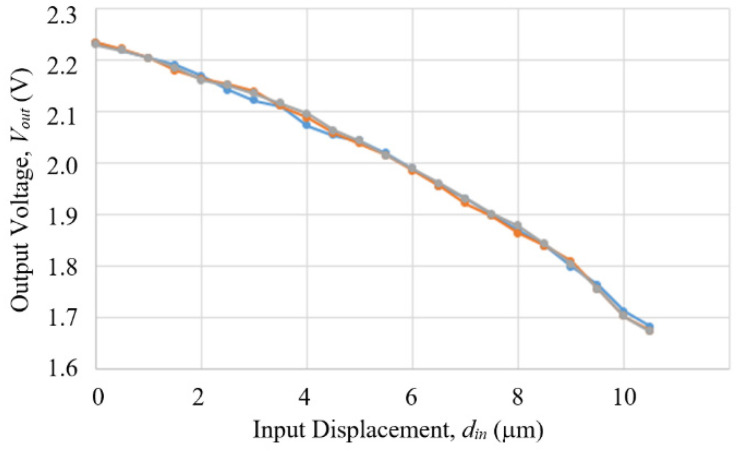
Results of three separate measurements of a prototype of the 25 mm compact model by using a Hall sensor and a tape magent to measure output displacement, increasing input displacement meaning decreasing gap distance between the Hall sensor and the tape magnet on the output arm.

**Figure 28 sensors-23-00326-f028:**
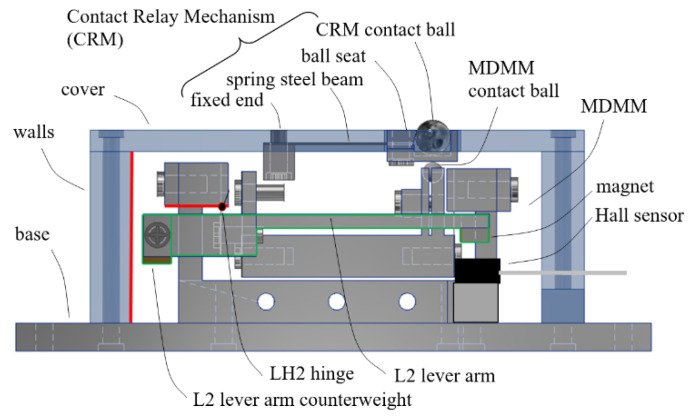
Design of the packaging and the Contact Relay Mechanism.

**Figure 29 sensors-23-00326-f029:**
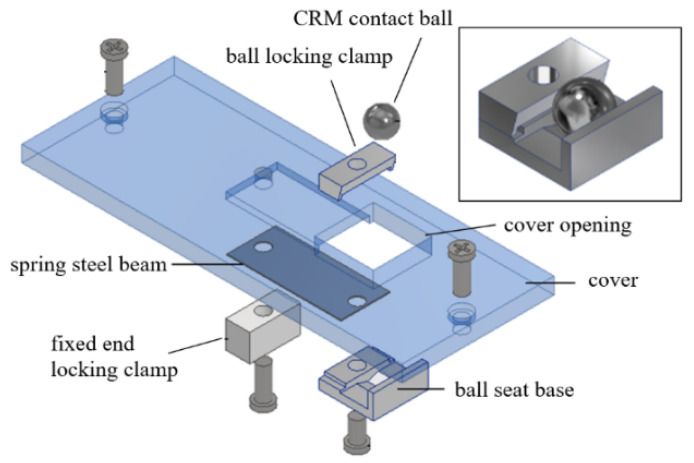
Design of the Contact Relay Mechanism in exploded view.

**Figure 30 sensors-23-00326-f030:**
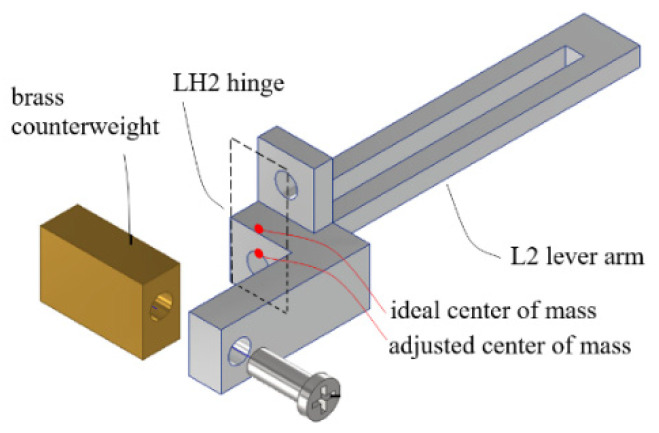
Design and construction of a balanced lever arm 2.

**Figure 31 sensors-23-00326-f031:**
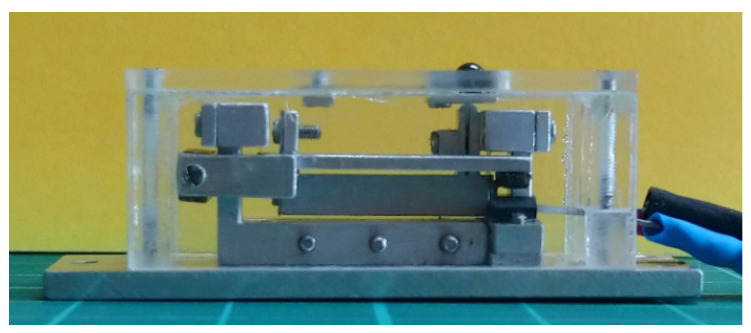
The completed LCMDS Alpha Model, in side view.

**Figure 32 sensors-23-00326-f032:**
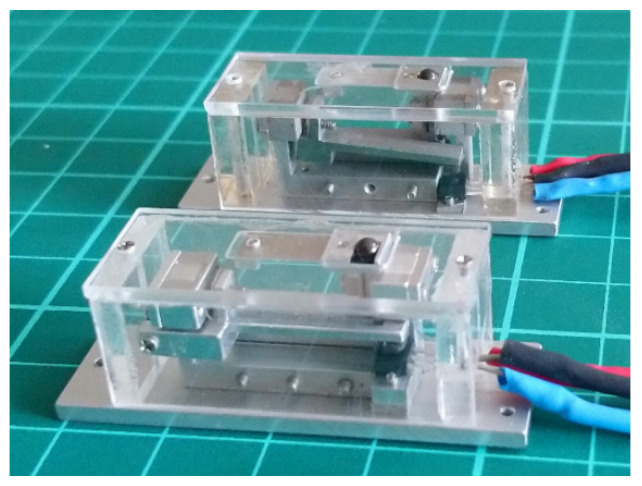
Two packaged LCMDS prototypes, the Alpha Model in the foreground.

**Figure 33 sensors-23-00326-f033:**
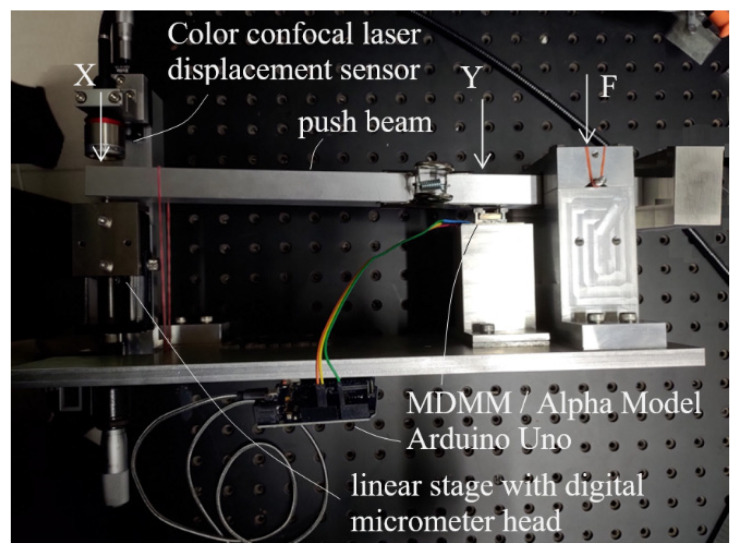
A second setup for testing/calibrating the LCMDS prototypes at different orientations, the picture showing the setup operating in horizontal orientation.

**Figure 34 sensors-23-00326-f034:**
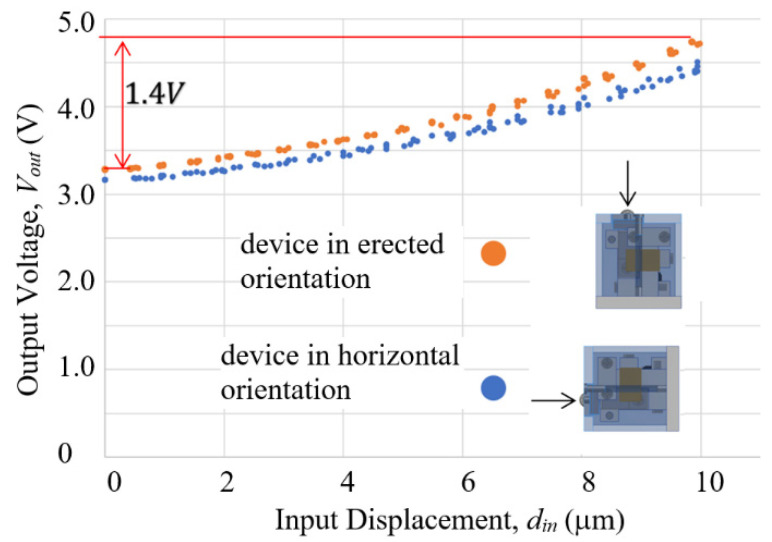
Results of calibrations of the LCMDS Alpha Model operating in two different orientaitons.

**Figure 35 sensors-23-00326-f035:**
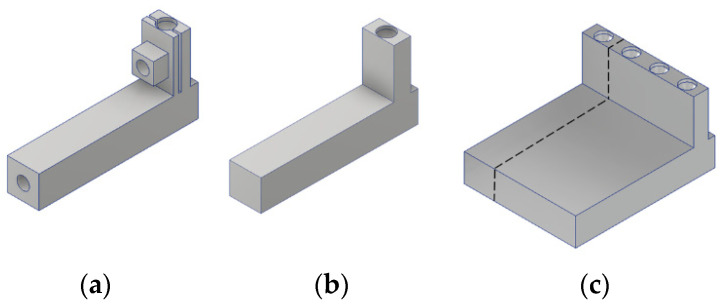
An example showing the idea of part shape simplification by extruded geometry, from (**a**) to (**b**) and manufacturing by applying a multi-part intermediate object (**c**).

**Figure 36 sensors-23-00326-f036:**
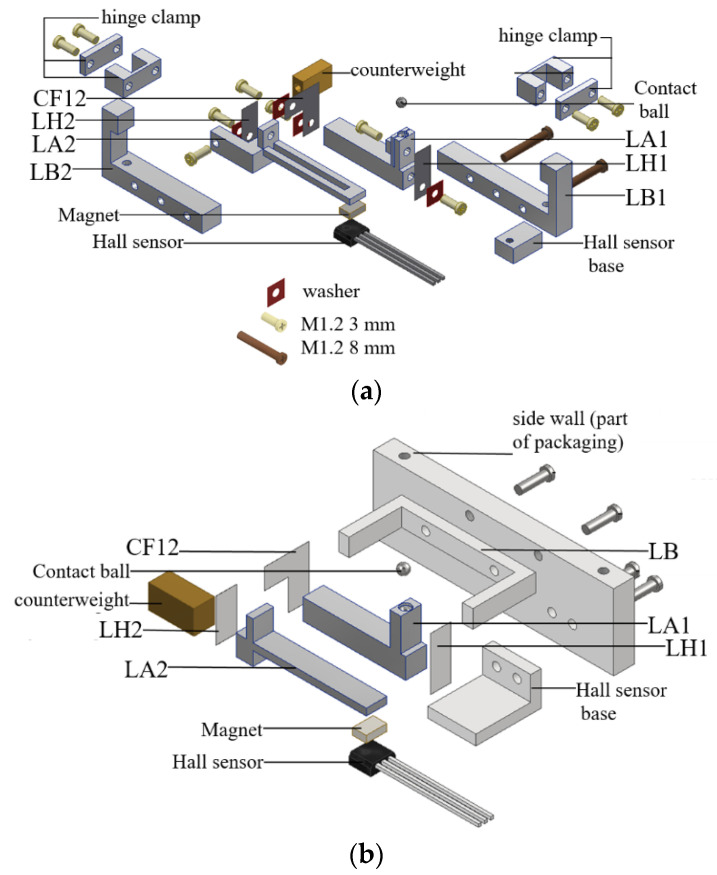
Design modification of the Alpha Model (excluding the packaging and the CRM) for small lot size production; (**a**) current Alpha Model and (**b**) after modification and simplification.

**Table 1 sensors-23-00326-t001:** Comparison of specifications and prices of typical commercially available displacement sensors.

Category	Range (mm)	Accuracy (μm)	Resolution (μm)	Dealer Price in USD *^a^* (Brand/Manufacturer)
Capacitive sensor	0.05	0.15	0.001	$2000 (με) *^b^*
Eddy current sensor	1	1	0.1	$2100 (Kaman) *^c^*
Inductive sensor (LVDT)	1	6	-- *^i^*	$650 (με) *^d^*
Laser triangulation	10	12	1	$1300 (με) *^e^*
Digital contact displacement sensor	12	1	0.1	$1600 (Keyence) *^f^*
Color confocal laser displacement sensor	7	0.94	0.25	>$8000 (Keyence) *^g^*
Hall sensor	0.25–2.5	0.1~1% of full scale	5	$5 *^h^*

*^a^* From quotations by dealers. *^b^* Micro Epsilon product specifications [1]. *^c^* Kaman product specifications [2]. *^d^* Micro Epsilon product specifications [3]. *^e^* Micro Epsilon product specifications [4]. *^f^* Typical model GT2-PK12A [5]. *^g^* Typical model CL-L030 [6]. *^h^* General specification, see Chap. 3 of [7]. *^i^* Depending on amplifying electronics.

**Table 2 sensors-23-00326-t002:** Relation between mechanical magnification ratio and measurement range, with corresponding accuracy and resolution, based on a Hall sensor of a measurement resolution of 5 μm and an accuracy of 0.1–1% of the full measurement range.

Measurement Range (μm)	Magnification Ratio	Magnified Measurement Range (mm)	Accuracy (μm)	Resolution (μm)
10	250	2.5	0.01–0.1	0.002–0.02
10	100	1	0.01–0.1	0.005–0.05
25	100	2.5	0.025–0.25	0.005–0.05
50	50	2.5	0.05–0.5	0.01–0.1
100	25	2.5	0.1–1	0.02–0.2
250	10	2.5	0.25–2.5	0.05–0.5
500	5	2.5	0.5–5	0.1–1

**Table 3 sensors-23-00326-t003:** Parameters of performance of the LCMDS Alpha Model.

Alpha Model	Repeatability (μm)	Accuracy (μm)	Resolution (μm)
Erected orientation	0.20	0.42	0.03
Horizontal orientation	0.54	0.36	0.04

## Data Availability

Not applicable.

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
