# Peer review of "Precision Low-Cost Compact Micro-Displacement Sensors Based on a New Arrangement of Cascaded Levers with Flexural Joints"

_sensors, 2022, doi:10.3390/s23010326_

Round 1

Reviewer 1 Report

The reviewer believes that the current version of the manuscript is not yet ready for publication; the authors are encouraged to consider the following comments and suggestion and revise the manuscript accordingly.

This research discusses the design and development of a new contact type compact micro-displacement sensor of sub-micron precision at a low cost. The fundamental concept of the new system is based on a mechanical magnifying mechanism to show the displacement at sub-micron level. The reviewer believes that the current version of the manuscript is not yet ready for publication; the authors are encouraged to consider the following comments and suggestion and revise the manuscript accordingly.

1. The authors should streamline the Abstract section. Currently, it is very short and does not cover all necessary information. The Abstract section should focus on explaining why the research is needed, what the research is about, what the methodology is, and what the conclusion is. Do not include any unnecessary information but the essential information must be provided.

2. The authors should consider reorganizing the manuscript to include the following sections: Introduction, Background, Methodology, Results and Discussion, and Conclusions. The Introduction section should focus on introducing the research objectives and stating the research questions that need to be answered, while the Background section should focus on reviewing of related literature and presenting the process of finding the research gap.

3. An extensive editing of the manuscript is required. Need to have a professional editor to go through any grammatical, spelling, and linguistic issues.

4. The authors should include a document to explain their algorithms. Such a document will assist researchers in replicating the proposed method. The authors also need to go through the equations to make sure all elements in the equations are denoted.

5. The conclusion section needs to be improved. How can a single sentence be a paragraph? The authors need to combine all the sentences to make a single paragraph conclusion section. In addition, many of the statements in the Conclusion section is redundant since they have been stated over and over again in the previous sections.

6. All figures and tables need to be improved. None of the figures is legible.

Reviewer 2 Report

The paper by Che-Chih Tsao et al. illustrates the design and the realization of a relatively complex micro-displacement sensor with sub-micron resolution. The design is illustrated in detail and the experimental results of the realized prototype are interesting.
The work was developed in a detailed way, and also simulations completed the design. This author appreciated the systematic and complete study.
The article appears useful for specialists in the sector.

I suggest only lightening some sentences and a moderate English revision is required, but fundamentally I recommend publishing the article in its current form.

Reviewer 3 Report

The research paper is well written. The actual problem of using a small displacement sensor is solved. The publication is very valuable from a technological point of view, as it is possible to apply the described methods and obtained results in practice (even on an industrial scale). The article is mature, it is a clear scientific innovation. Taking into account all the circumstances, I propose to accept the publication without correction.

Reviewer 4 Report

1. The assembly components and process of the sensor seem to be relatively complex. Whether the assembly accuracy has a great impact on its measurement accuracy can be properly described in this paper.

2. Many variable symbols in the text need to be italicized. The curve of each color in Figure 27 does not give a specific meaning.

3. The unsymmetrical lever mechanism used in the flexible mechanism will produce large parasitic displacement. How does this displacement affect the measurement accuracy of the sensor.

4. The flexible mechanism in the sensor adopts multi-stage series connection, which will reduce the dynamic characteristics of the sensor. This is not fully explained in the paper, nor is the dynamic performance of the sensor tested.

5. The AD conversion of the sensor has only 10 bits. Can you use a higher bit, such as a 24 bit AD converter to further improve the accuracy and resolution of the sensor.

6. What is the potential application scenario of the sensor designed in this paper? As far as I know, the current nano displacement measurement in piezoelectric driven nanopositioning platform mainly depends on the method of strain gauge SGS, with low cost and high resolution. Whether it is possible to add this part of the discussion.

Round 2

Reviewer 1 Report

The authors have address all my comments.

Reviewer 4 Report

This article has done detailed work and is worth publishing